# Anti-Aging Effects of Anthocyanin Extracts of *Sambucus canadensis* Caused by Targeting Mitochondrial-Induced Oxidative Stress

**DOI:** 10.3390/ijms24021528

**Published:** 2023-01-12

**Authors:** Xiaoqing Hu, Yimeng Yang, Shi Tang, Qiuyan Chen, Meiyu Zhang, Jiaoyan Ma, Jianchun Qin, Huimei Yu

**Affiliations:** 1Key Laboratory of Pathobiology, Ministry of Education, Department of Pathophysiology, College of Basic Medical Sciences, Jilin University, Changchun 130021, China; 2College of Plant Science, Jilin University, Changchun 130062, China

**Keywords:** anthocyanin, cell senescence, PI3K/AKT/mTOR pathway, autophagy, apoptosis

## Abstract

Anthocyanin is a natural antioxidant agent extracted from the fruits of *Sambucus canadensis*, which has been considered to have potential anti-aging effects. Cell senescence is the primary cause of aging and related diseases. Recently, research on the development of compounds for eliminating senescent cells or damaged organs have shown prospects. The compounds which promote the clearing of senescent cells are called “senolytics”. Though anthocyanin is considered to have potential anti-aging effects owing to its anti-inflammatory and antioxidant activities, the mechanism of the elimination of senescent cells remains unclear. In this study, we prepared anthocyanins extracted from the fruits of *Sambucus canadensis* and evaluated their anti-aging effects in vivo and in vitro. We found that anthocyanin could significantly reduce cell senescence and aging of the lens by inhibiting the activity of the PI3K/AKT/mTOR signaling pathway, consequently promoting the apoptosis of senescent cells, increasing the autophagic and mitophagic flux, and enhancing the renewal of mitochondria and the cell to maintain cellular homeostasis, leading to attenuating aging. Therefore, our study provided a basis for anthocyanin to be used as new “senolytics” in anti-aging.

## 1. Introduction

With medical advancements, the average life expectancy is also increasing. There has been a continuous increase in the morbidity of age-related diseases with the rapidly growing elderly population [1,2]. Elderly people are commonly plagued by age-related cataracts, sarcopenia, and other diseases, which cause visual impairment and mobility problems and lead to a decline in quality of life all over the world [3]. Understanding the mechanism of cell senescence is an important means to delay senescence, explore the drug mechanism of the prevention and treatment of age-related diseases, and improve the quality of life of older adults.

Recently, it has been shown that the accumulation of senescent cells is related to age-related diseases, such as cataracts, Alzheimer’s disease, atherosclerosis, osteoarthritis, type 2 diabetes, and others [4,5,6]. Our bodies need to constantly create new cells to replace senescent and damaged cells in order to maintain our health. In 2015, Kirkland first found that dasatinib combined with quercetin could selectively induce the apoptosis of senescent adipocytes and endothelial cells and reduce the age-related degeneration of tissues and organs. They defined these drugs as “senolytics” [7]. Therefore, the use of senolytics has become a promising method of delaying cellular senescence. Apoptosis and autophagy pathways play important roles in the clearance of senescent cells and damaged organs [8,9]. Exploring the mechanism of senolytics by targeting apoptosis and autophagy will provide the basis for the development of strategies for anti-aging treatment and pharmaceutical advancements.

Some natural compounds have anti-aging potential. Anthocyanins are water-soluble flavonoid pigments extracted from the fruits of *Sambucus canadensis*, or elderberry, and have anti-inflammatory and antioxidant biological properties [10]. Owing to the polyphenolic structure of anthocyanin, they can react with metal ions, prevent the catalytic effect of active metal ions, inhibit the generation of free radicals, and produce antioxidant effects [11]. Studies suggest that anthocyanins have antioxidant activity and can play a role in anti-aging and anti-tumor treatments [12,13]. However, whether anthocyanin is suitable as a “senolytic” and what the targets of anthocyanin are in senescent cells are worth exploring thoroughly.

It has been reported that PI3K/AKT/mTOR signaling is a crucial pathway that regulates cell senescence [14]. Though some studies have suggested that cell senescence could be prevented through the activation of the PI3K/AKT pathway [15], in the field of aging research, the inhibition of mTOR has been regarded as a major discovery [16]. It has been found that the inhibition of the mTOR by genetic or pharmacological intervention extends the lifespan of invertebrates, including yeast, nematodes, fruit flies, and mice [17,18]. However, the underlying mechanism is still unclear. Considering that PI3K/AKT/mTOR achieve signaling balance homeostasis through degradation pathways such as autophagy and apoptosis, this study aims to understand the mechanism of the clearing of senescent cells and the exploration of targets of anthocyanin through taking the PI3K/AKT/mTOR pathway as the entry point.

In this study, the models of senescent cells and aging animals were induced by D-galactose as the research targets and the effects of anthocyanin extracted from the fruits of *Sambucus canadensis* on PI3K/AKT/mTOR pathway activity, apoptosis, autophagy, and mitophagy pathways in vivo and in vitro were observed to explore the mechanism of cell senescence and the effects of anthocyanin as a senolytic.

## 2. Results

### 2.1. Identification of Anthocyanin

1H NMR (400 MHz, Methanol-d4) δ 8.36 (s, 1H), 7.42 (d, J = 7.5 Hz, 1H), 7.31 (s, 1H), 7.10 (d, J = 3.6 Hz, 1H), 7.02 (dd, J = 5.4, 2.2 Hz, 1H), 6.96 − 6.69 (m, 4H), 6.53 (d, J = 21.4 Hz, 1H), 6.20 (d, J = 43.0 Hz, 1H), 5.98 − 5.90 (m, 1H), 5.10 (d, J = 3.7 Hz, 1H), 4.72 − 4.63 (m, 1H), 4.57 (d, J = 9.9 Hz, 1H), 4.47 (d, J = 7.8 Hz, 1H), 4.27 (d, J = 20.1 Hz, 1H), 3.94 (s, 1H), 3.89 (s, 1H), 3.86 (d, J = 1.6 Hz, 1H), 3.83 (d, J = 2.2 Hz, 1H), 3.79 (t, J = 2.7 Hz, 1H), 3.77 (d, J = 2.8 Hz, 1H), 3.70 (d, J = 5.9 Hz, 1H), 3.67 − 3.62 (m, 2H), 3.62 − 3.40 (m, 6H), 3.36 (d, J = 3.5 Hz, 3H), 3.34 (dd, J = 3.8, 2.3 Hz, 2H), 3.30 − 3.25 (m, 2H), 3.15 − 3.09 (m, 1H). The anthocyanin (70.4%, consistency of Johansen’s data) was identified as cyanidin 3-O-[6-O-(E-p-coumaroyl-2-O-(β-D-xylopyranosyl)-β-D-glucopyranoside)]-5-O-β-D-glucopyranoside (Figure 1).

### 2.2. Effects of Anthocyanin on General Physiological Indexes and Cell Senescence-Associated Phenotypes in an Aging Mouse Model Induced by D-Galactose

Prior to D-galactose treatment initiation, the coat color of each group was well-distributed and glossy, both calls of nature were regular, the mental status was normal, and the sensitive responses worked smoothly. However, post-treatment with D-galactose for 9 weeks, the mice in the D-galactose(D-gal) group showed easy depilation, a slanted dark coat color, slower movement, mental fatigue, and weight gain (Figure 2A). Hemal biochemistry assays exhibited increased levels of glucose and aspartate aminotransferase and decreased albumin and total protein (Table 1), indicating signs of aging in the mice in the model group. Compared with the D-gal group, mice in the anthocyanin and vitamin E (VE) groups had a glossier coat color, their calls of nature were regular, their sensitive responses worked smoothly, and weight loss was observed. Hematologic biochemistry assays showed that glucose and aspartate aminotransferase levels decreased, and the albumin and total protein levels decreased (Table 1). These results demonstrate that D-galactose could induce signs of aging in mice while anthocyanin (especially in the low and middle groups) and vitamin E could slow down the same.

Aging of the lens, which leads to decreased transparency, is a clear feature of age-related cataracts and is one of the signs of aging. Hence, we used the lens tissue of mice to observe signs of aging. Staining with HE revealed degeneration of the anterior lens epithelial cells and the presence of vacuoles under the anterior capsule in the mice treated with D-galactose. However, in the anthocyanin and vitamin E groups, the anterior lens epithelial cells were arranged in an orderly way, and no significant pathological changes were observed (Figure 2B). Cell cycle arrest is one of the main features of cell senescence [19], and we evaluated aging in mice by detecting the cell cycle. By detecting the expression of p53 and p21, we found that the protein expression of p53 and p21 in the lens of aging mice was increased, but the same was decreased in the anthocyanin and vitamin E groups (Figure 2C,D). Therefore, the results show lens senescence in the model group, while in the anthocyanin group, reduced cell cycle arrest in senescent tissue played a role in delaying senescence.

In view of the above results, we further explored the molecular mechanism of cell senescence in vitro and the mechanism by which anthocyanins reverse cell senescence.

### 2.3. Anthocyanin Attenuated Cell Senescence Induced by D-Galactose

The cell proliferation experiment using the MTT assay showed that D-galactose had no significant effect on cell viability at 40 mM concentration (Figure 3A). However, SA-β-gal activity was significantly enhanced when treated with D-galactose, indicating an increased ratio of senescent cells (Figure 3B,C). Furthermore, by detecting the mRNA expression of p53, p21WAF1, and p16INK4A, the result showed that their levels were increased in cells induced by D-galactose, whereas anthocyanin was downregulated the same. This suggested that anthocyanin could inhibit cell cycle arrest induced by D-galactose. The above results indicated that senescent cells increased continuously when treated with D-galactose and that their accumulation decreased the renewal of normal cells. They also showed that anthocyanin could reduce D-galactose-induced cell senescence.

### 2.4. Anthocyanin Inhibited the PI3K/Akt/mTOR Signaling Pathway of Senescent Cells

The further study explored the molecular mechanisms underlying cellular senescence. It was found that the PI3K/AKT/mTOR signaling pathway regulates cell growth, apoptosis, autophagy, and other pathways closely related to the regulation of cell senescence. In this study, we found that the protein expression levels of p-AKT/AKT and p-4EBP1/4EBP1 in the model group did not increase significantly (Figure 4A,B). However, in the group treated with anthocyanin, the protein expression levels of p-AKT/AKT and p-4EBP1/4EBP1 decreased dramatically (Figure 4A,B). This indicated that anthocyanin inhibited the PI3K/AKT/mTOR pathway in the D-galactose-induced senescent cells.

### 2.5. Anthocyanin Increased Apoptosis of Senescent Cells Which Were Induced by D-Galactose

Due to the increased activity of PI3K/AKT/mTOR, cells became resistant to apoptosis and prone to senescence. We measured the apoptosis of senescent cells with or without anthocyanin. By detecting the expression of apoptosis-related proteins, it was shown that the ratios of Bax/Bcl-2 and Bak/Mcl-1 were not significantly increased in the model group, indicating that little apoptosis occurred in senescent cells (Figure 5A,B). The ratios of Bax/Bcl-2 and Bak/Mcl-1 increased in the cells treated with anthocyanin, demonstrating that it induced apoptosis in senescent cells (Figure 5A,B). These results suggested that anthocyanin might promote the clearance of senescent cells by increasing apoptosis and the proportion of healthy cells.

### 2.6. Anthocyanin Increased Autophagic Flux in Senescent Cells Induced by D-Galactose

Autophagy may also be one of the mechanisms responsible for the clearance of damaged organelles to attenuate cell senescence. The PI3K/AKT/mTOR signaling pathway is a major pathway regulating autophagy. Autophagic flux is enhanced when the PI3K/AKT/mTOR pathway is inhibited. In this study, we investigated whether anthocyanin could clear damaged organelles by enhancing autophagic flux. To this end, autophagic flux was evaluated in the cells treated with D-galactose alone or in combination with anthocyanin. By detecting the expression of autophagy-related proteins p62 and LC3-II/I, the result showed that the expression of p62 did not increase significantly, and the ratio of LC3-II/I increased in the model group, suggesting that its accumulation might be due to blocked autophagic flux in senescent cells. The expression of p62 and LC3-II/I was decreased in the anthocyanin-treated group, indicating that anthocyanin might promote autophagy and enhance autophagic flux (Figure 6A–C). To further evaluate autophagic flux in cells, the expression of autophagy-related proteins p62 and LC3-II/I was detected when the cells were treated with the lysosomal inhibitor chloroquine (CQ). As shown in Figure 6A–C, the expression of LC3-II/I proteins significantly accumulated when autophagic flux was blocked by CQ in the anthocyanin group, while the LC3-II/I protein expression did not accumulate obviously in cells treated by D-galactose. This further confirmed that autophagic flux in the senescent cells was blocked, and that anthocyanin enhanced it.

### 2.7. Anthocyanin Improved the Mitochondrial Function of Senescent Cells

Mitochondria play an important role in supplying cellular energy and maintaining redox homeostasis [20]; therefore, they are essential for delaying cell senescence. Damaged mitochondria can be cleared by mitophagy to maintain homeostasis [21]. In this section, mitochondrial function was evaluated through mitophagy, mitochondrial biosynthesis, and Adenosine triphosphate (ATP) production in senescent cells. As shown in Figure 7A,B, the expression of mitophagy-related proteins PINK1 and Parkin increased in the model group, while the ATP content in the cells and the mitochondrial copy number decreased (Figure 7F,G). This suggested that the number of mitochondria and the production of ATP decreased in senescent cells, and that mitophagic flux might have been enhanced or blocked. In cells treated with anthocyanin, the expression of PINK1 increased slightly, while that of Parkin decreased (Figure 7A,B), and the mitochondrial copy number and ATP content increased (Figure 7F,G). Mitophagic flux was tested to confirm whether the decreased expression of Parkin was caused by degradation through mitophagy. When mitophagic flux was blocked by chloroquine (CQ), the expression of Parkin and PINK1 increased significantly when cells were treated with anthocyanin, while there was little change in the model group (Figure 7C–E). It was further observed that mitophagic flux was blocked in senescent cells, and that anthocyanin increased it and improved mitochondrial function in the same.

### 2.8. Verification of the Mechanism of Anthocyanin in Delaying the Aging of Mouse Lens

To verify the results in vitro, the molecular mechanisms related to aging were further examined in an animal model of aging using lenses that are susceptible to age-related lesions as research objects to explore the molecular mechanisms of cell senescence. First, the protein expression levels of p-AKT/AKT and p-4EBP1/4EBP1 increased in the lens of the model group (Figure 8A,B), demonstrating that the activity of the PI3K/AKT/mTOR signaling pathway was enhanced. According to the measurement of apoptosis-related proteins, the ratio of Bax/Bcl-XL, Bak/Mcl-1, and the expression of cleaved caspase-3 decreased (Figure 8C,D), indicating a reduction in apoptosis in aging mice lenses. When mice were treated with anthocyanin, the activity of the PI3K/AKT/mTOR signaling pathway decreased (Figure 8A,B). The ratio of Bax/Bcl-XL, Bak/Mcl-1, and expression of cleaved caspase-3 increased (Figure 8C,D), indicating that anthocyanin promoted apoptosis in aging mice lenses.

In the measurement of autophagy, the expression of p62 and LC3II/I increased in the model group, whereas that of p62 decreased. The expression of LC3II/I increased in the anthocyanin-treated group (Figure 9A,C,D). Since it is difficult to evaluate autophagic flux in vivo, the mRNA expression of p62 was measured to determine whether the decreased expression of p62 was caused by reduced synthesis or increased degradation through autophagy. As shown in Figure 9B, when the cells were treated with anthocyanin, the mRNA expression of p62 significantly increased, suggesting that the decreased expression of p62 was due to increased degradation through increased autophagic flux, further indicating that anthocyanin increased autophagic flux in aging mice lenses.

## 3. Discussion

Cell senescence is a major risk factor for many chronic diseases [22]. Senescent cells reveal irreversible growth arrest and resistance to apoptosis [23], therefore, the cell renewal is blocked, and the regenerative and reparative abilities of tissues are decreased. In order to maintain normal physiological, biochemical, and immune functions, the organisms need to induce the elimination of senescent cells, accelerating cell renewal. It is of great significance to find “senolytics” that can eliminate senescent cells and elucidate the mechanism of the prevention and treatment of age-related disease [24].

In-depth research into the signaling pathway of regulating cell senescence is helpful to elucidate the molecular mechanism of cell senescence and to develop therapeutic targets for senolytics. In this study, it was found that anthocyanins, as compounds with antioxidant activity, could reverse the senescent phenotypes of mice, especially those in the low- and middle-dose groups (50 mg/kg/d and 100 mg/kg/d anthocyanin), and we further explored the molecular mechanism in vitro. Previous studies have found that the PI3K/AKT/mTOR pathway is closely related to cell senescence [25]. In view of the central role of the PI3K/AKT/mTOR pathway in regulating cell proliferation, apoptosis, and autophagy, in this study, we evaluated the activity of the PI3K/AKT/mTOR pathway in senescent cells and tissues, and further elucidated the mechanism by which anthocyanin regulates cell senescence through the cell cycle, apoptosis, autophagy, and mitophagy based on inhibiting the PI3K/AKT/mTOR pathway.

It was found in our study that the activity of the PI3K/AKT/mTOR pathway was not enhanced significantly in senescent cells which were induced by D-galactose (Figure 4D). Meanwhile, the expression of p-Akt and p-4EBP1 were also increased in the lens and muscle of aging mice (in other research by our team), it indicated that the activity of the PI3K/AKT/mTOR pathway increased with aging, and, with the presence of stress, cells regulated the PI3K/AKT/mTOR pathway to increase resistance to stress. The arrest of the cell cycle is one of the key features of cell senescence, and the activated PI3K/AKT/mTOR signal enhances the p53/p21 or p16 pathway and inhibits the cell cycle by upregulating the translation of p53 [26]; therefore, the protein expression and the mRNA expression of p53 increased remarkably (Figure 2C). Furthermore, the PI3K/AKT/mTOR pathway leads to resistance to apoptosis and a reduced capacity for degradation. It increases the apoptotic activity through the phosphorylation of pro-apoptotic proteins such as Bax by Akt [27,28], or inhibits the degradation of these anti-apoptotic proteins through the proteasome or autophagy pathway by inducing the phosphorylation of XIAP and Mcl-1 [29,30], then increases the expression of anti-apoptotic proteins. In addition, the inhibition of the PI3K/AKT/mTOR pathway activates autophagy by increasing the activity of the ULK1-FIP200-ATG13 complex [31], and enhanced autophagy could promote the clearance of damaged proteins or organelles, an increased immune response, and a reduced inflammatory response. It suggests that the PI3K/AKT/mTOR pathway may be an important target for anthocyanins in reversing cell senescence.

In recent years, a multifaceted study of nanoscale complexes in medicine, including antioxidant nanoparticles, especially natural antioxidant nanoparticles, has become important in view of a number of positive features that make it possible to create new drugs based on them, significantly improving the properties of existing drugs or improving the solubility of drugs.

In this study, anthocyanin increased the ratio of pro-apoptotic to anti-apoptotic proteins in senescent cells/tissues, suggesting that the upregulation of apoptotic pathways in senescent cells played a key role in the clearance of senescent cells. In addition, as an important clearance pathway, autophagic flux also decreased significantly in senescent cells. In vivo, the protein expressions of p62 and LC3II/I were increased in the lens and muscle in the mice model, while the mRNA level of p62 was not increased, indicating decreased autophagic flux in senescent cells. However, when treated with anthocyanin, the mRNA expression of p62 increased, while the protein expression decreased, suggesting that anthocyanin activated the degradation pathway by increasing autophagic flux in senescent cells, and eliminated damaged proteins and organelles.

Mitochondrial dysfunction leads to defects in ATP synthesis, which is one of the important reasons for inducing cell senescence [32]. In eukaryotes, it can reduce the production of reactive oxygen species (ROS) and maintain mitochondrial function by synthesizing new and removing damaged mitochondria through mitochondrial quality control, including mitochondrial biosynthesis, fusion and fission, and mitophagy [33]. It was observed in our study that anthocyanin could increase the mitochondrial copy number, promote mitochondrial biosynthesis, increase ATP content in cells, enhance the anti-oxidant capacity, and improve the proportion of healthy mitochondria. In the beginning of mitophagy, the mitochondria network was broken through mitochondrial fission, so as to facilitate the entry of mitochondria into autophagic bodies. Our results showed that the expression of mitophagy-related proteins PINK1 and Parkin increased, suggested that mitophagy was activated. In addition, mitophagy is an important pathway for eliminating damaged organelles.

## 4. Materials and Methods

### 4.1. Extraction and Isolation of Anthocyanin

Ripe fruits of *S. canadensis* were air dried outside and extracted 3 times with MeOH containing 0.03% HCl at room temperature. After filtration and evaporation, the obtained crude extract was suspended in H_2_O and extracted with EtOAc to afford fraction A (100.0 g). Fraction A was isolated with silica gel column and eluted with CH2Cl2-MeOH (1:1) to give the anthocyanin fraction (8.0 g). This fraction was subjected to MPLC (CHEETAH II, Model CH-20 4P, Agela Technologies) with C18 spherical 20–35 um 100 A gel column eluted with 30% MeOH in H_2_O to 100% MeOH to give fractions 1–3. Fraction 2 was purified by HPLC (ultimate 3000) with C18 -5μm column using CF3COOH-MeOH-H2O (1:5:4) to give a compound.

### 4.2. Animals

In total, 36 male Kunming mice of specific pathogen-free (SPF) grade (25–30 g) were purchased from Changchun Yisi Experimental Animal Technology (Changchun, China). The mice were kept at a constant temperature (22 °C), with a light/dark cycle of 12 h. After one week adaptation the rats were divided into 5 groups and each group had 6 mice: Control group(0.9% saline), model group (500 mg/kg/d D-gal), low dose of anthocyanin group (500 mg/kg/d D-gal + 50 mg/kg/d anthocyanin), middle dose of anthocyanin group(500 mg/kg/d D-gal + 100 mg/kg/d anthocyanin), high dose of anthocyanin group(500 mg/kg/d D-gal + 200 mg/kg/d anthocyanin), VE group, the mice were given these compounds for 8 weeks and weighed weekly. All animal experiments were performed in accordance with the National Guidelines for Experimental Animal Welfare, with approval from the Animal Welfare and Research Ethics Committee at Jilin University (Approval No. 2020–35) (Changchun, China).

### 4.3. Cell Lines and Cell Culture

Human embryonic kidney cells (HEK293T) were purchased from American Tissue Culture Collection (Rockville, MD, USA) and grown in DMEM medium (Gibco Life Technologies, Carlsbad, CA, USA), and supplemented with 10% fetal bovine serum (Invitrogen, Carlsbad, CA, USA) at 37 °C, 5% CO_2_ concentration.

### 4.4. Reagents and Antibodies

D-galactose was purchased from Shanghai Yuan Ye Bio-Technology Co., Ltd. (Shanghai, China), MTT [3-(4,5-Dimethyl-2-thiazolyl)-2,5-diphenyl-2H-tetra-zolium bromide] was purchased from Sigma-Aldrich (St. Louis, MO, USA). The antibodies used in this study including antibodies against Bcl-2, Bax, Bak, p62, LC3 I/II were purchased from Abcam (Cambridge, MA, USA). Antibodies against Cleaved caspase 3, Mcl-1, Pink, Parkin, Beta-actin, HRP-linked secondary antibody were purchased from Proteintech (Chicago, IL, USA). Antibodies against AKT, p-AKT, 4EBP1, p-4EBP1, p70s6k, p-p70s6k were purchased from Cell Signaling Technology (Boston, MA, USA).

### 4.5. Cell Viability Assay

Cell viability was measured using the 3-(4,5-dimethylthiazol-2-yl)-2,5-diphenyltetrazolium bromide (MTT) assays. HEK293T cells were seeded in 96-well plates at a density of 2500 cells/well. Then the cells were incubated in culture medium until 70–80% confluency. The cells were treated with D-galactose and/or Anthocyanin according to the experiments. After 72 h, the cells were incubated in the dark with MTT reagent (0.5 mg/mL) at 37 °C for 2 h. Following this, the medium was removed, formazan was dissolved in DMSO, and absorbance values were measured at 490 nm using a Vmax Microplate Reader (Molecular Devices, LLC, Sunnyvale, CA, USA).

### 4.6. Reverse Transcription-Polymerase Chain Reaction (RT-qPCR)

Total cellular RNA was extracted using TRIzol™ reagent (Invitrogen; Thermo Fisher Scientific, Inc., Waltham, MA, USA), cDNA was obtained by reverse transcription with PrimeScript RT Reagent Kit (TransGen Biotech, Beijing, China). For quantitative PCR analysis, cDNA was amplified using SYBR Green PCR Master Mix (TransGen Biotech Co. Ltd. Beijing, China) on Bio-Rad CFX (Applied Biosystems). Each sample was tested in triplicate. The ΔΔCT method was employed to determine the fold change in gene expression level. Melting curves were used to verify specificity of real-time PCR.

### 4.7. Histology

For histological analysis, the lens was further dehydrated through graded alcohol, cleared in xylene, and embedded in paraffin. Sections were cut to 4 μm for hematoxylin and eosin staining. Images were collected with a ZEISS (Imager Z2) laser microscope.

### 4.8. Western Blot

Cells were lysated in RIPA buffer with protease inhibitors. Lysates were cleared by centrifugation at 10,000× *g* for 15 min at 4 °C, boiled in loading buffer and resolved using SDS-PAGE. Proteins were transferred to polyvinylidene fluoride (PVDF) membranes and membranes were blocked with 5% milk, followed by incubation with primary antibodies overnight at 4 °C. Membranes were then incubated with HRP-conjugated secondary antibodies (Proteintech group, Chicago, IL, USA). ECL reagent (Thermo Fisher Scientific, Waltham, MA, USA) was used for immune-detection and visualization using Syngene Bio Imaging (Synoptics, Cambridge, UK).

### 4.9. ATP Measurements

The level of ATP in cells was measured using ATP Bioluminescence Assay Kit (Beyotime Technology, Nantong, China). Briefly, cells were lysed with a lysis buffer, followed by centrifugation (10,000× *g*, 2 min) at 4 °C. Then, the supernatant (10 μL) was mixed with 100 μL luciferase reagent. The emitted light was measured using a luminometer (BMG LABTECH Omage, GER, Ortenberg, Germany) to evaluate the level of ATP. Counts were normalized to protein concentration.

### 4.10. Determination of the Relative mtDNA Copy Number

Total cellular DNA was extracted using the TIANamp Genomic DNA kit (Tiangen Biotech Co., Ltd., Beijing, China). The relative mitochondrial DNA content was determined by RT-qPCR. DNA was amplified using SYBR Green PCR Master Mix (TransGen Biotech Co. Ltd.) for measuring the mitochondrial-encoded nicotinamide adenine dinucleotide dehydrogenase 1 (ND1) relative to nuclear-encoded gene 18S rRNA (18S). Each sample was tested in triplicate. The relative mtDNA copy number was calculated as the ratio of the level of amplification obtained for ND1, vs. 18S for each sample, and was normalized to the control group.

### 4.11. Data Analysis

All results are based on at least three independent experiments. All data are given as mean values standard deviation (S.D). Results between the two groups are compared using Student’s *t*-test. *p* < 0.05 was considered statistically significant difference, and *p* < 0.01 was considered extremely significant. Statistical analysis was performed with GraphPad Prism 8.0 (La Jolla, CA, USA).

## 5. Conclusions

Anthocyanin could significantly reduce cell senescence and the aging of the lens by inhibiting the activity of the PI3K/AKT/mTOR signaling pathway, consequently promoting the apoptosis of senescent cells, increasing autophagic and mitophagic flux, and enhancing the renewal of the mitochondria and the cell to maintain cellular homeostasis, leading to attenuating aging in vivo and in vitro (Figure 10).

## Figures and Tables

**Figure 1 ijms-24-01528-f001:**
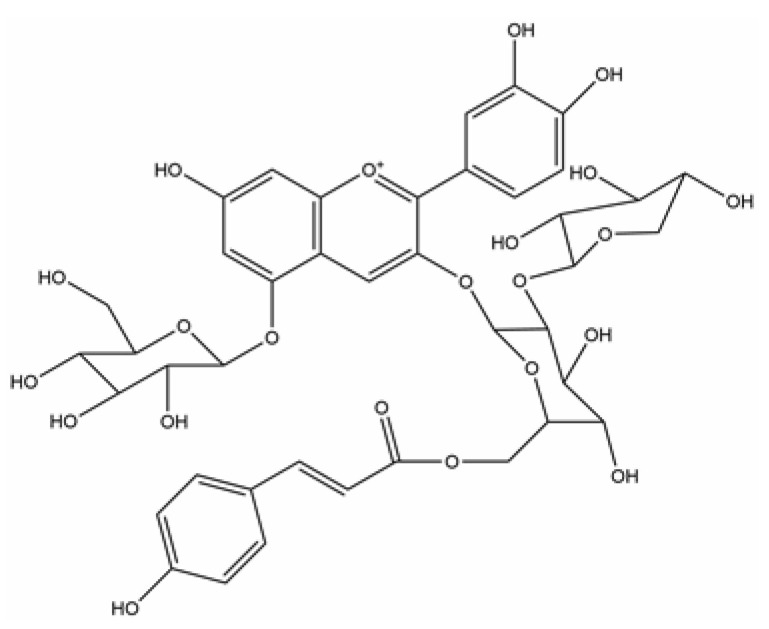
The chemical structure of anthocyanin.

**Figure 2 ijms-24-01528-f002:**
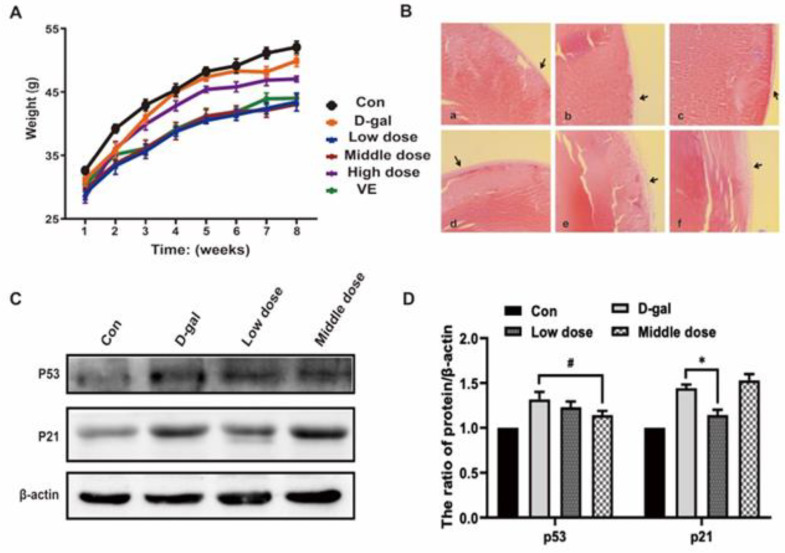
Effects of anthocyanin on general physiological indexes and cell senescence-associated phenotypes in an aging mouse model induced by D-galactose. (**A**) The body weights of mice induced by D-galactose alone or combination of anthocyanin/vitamin E. (**B**) Histological sections of mouse lens. (a–f) are groups divided according to the following: (a) Control group. (b) D-galactose group. (c) Low-dose anthocyanin group. (d) Middle-dose anthocyanin group. (e) High-dose anthocyanin group. (f) Vitamin E group. (Original magnification: ×100; Arrows: Lens epithelial cells.) (**C**,**D**) The expressions of p53 and p21 were measured using western blot. Data are presented as mean ± SD, n = 3, * *p* < 0.05, ^#^
*p*< 0.05.

**Figure 3 ijms-24-01528-f003:**
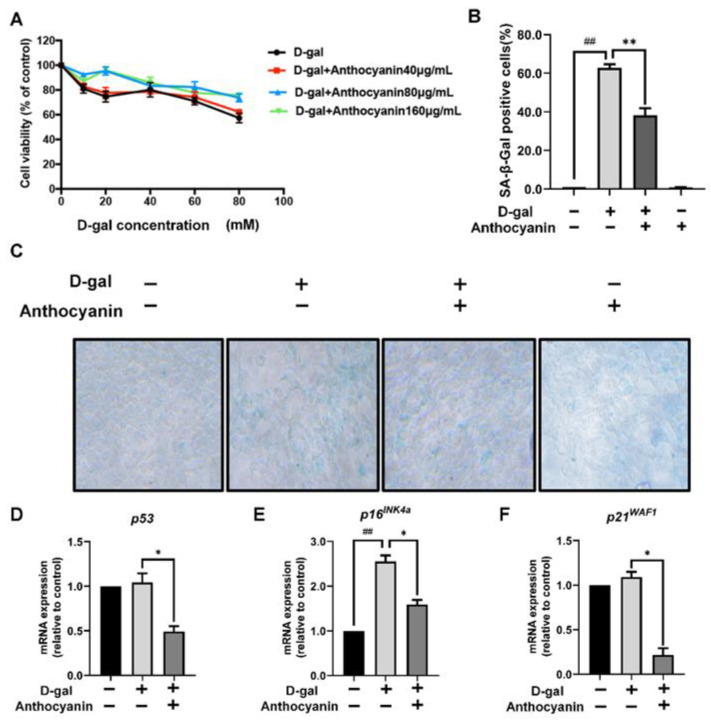
Anthocyanin-attenuated cell senescence induced by D-galactose. (**A**) Cells were treated with 0–80 mM D-galactose alone or in combination with 40–160 μg/mL anthocyanin for 72 h, and cell viability was measured by MTT assay. (**B**,**C**) Cells were treated with 40 mM D-galactose alone or in combination with 80 μg/mL anthocyanin for 72 h, and the activity of senescence-associated β-galactosidase (SA-β-gal) was observed and photographed using optical microscope (Original magnification: ×200) and the quantification of SA-β-gal positive cells was shown in (**B**). (**D**–**F**) Cells were treated with 40 mM D-galactose alone or in combination with 80 μg/mL anthocyanin for 72 h, and the mRNA expression of p53, p21WAF1, and p16INK4a were measured by RT-qPCR. Data are presented as mean ± SD, n = 3. * *p* < 0.05, ** *p* < 0.01, ^##^
*p* < 0.01.

**Figure 4 ijms-24-01528-f004:**
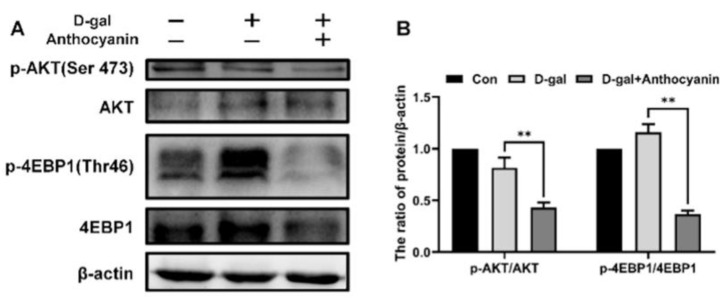
Anthocyanin inhibited the PI3K/Akt/mTOR signaling pathway of senescent cells. (**A**,**B**) Cells were treated with 40 mM D-galactose alone or in combination with 40 μg/mL anthocyanin for 72 h, and the expression of p-AKT/AKT, p-4EBP1/4EBP1 were measured by western blot. Data are presented as mean ± SD, n = 3. ** *p* < 0.01.

**Figure 5 ijms-24-01528-f005:**
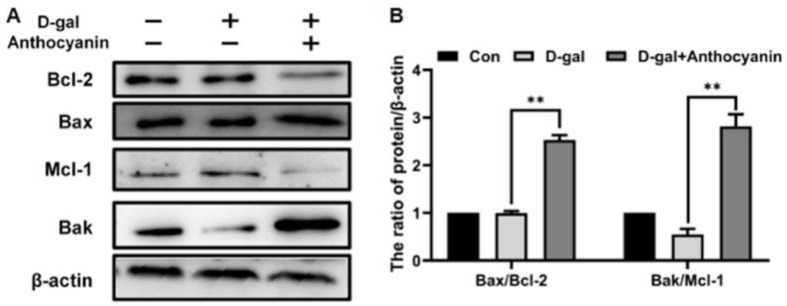
Anthocyanin inhibited the PI3K/Akt/mTOR signaling pathway of senescent cells. (**A**,**B**) Cells were treated with 40 mM D-galactose alone or in combination with 80 μg/mL anthocyanin for 72 h. The expression of Bcl-2 family proteins was measured using western blot. Data are presented as mean ± SD, n = 3. ** *p* < 0.01.

**Figure 6 ijms-24-01528-f006:**
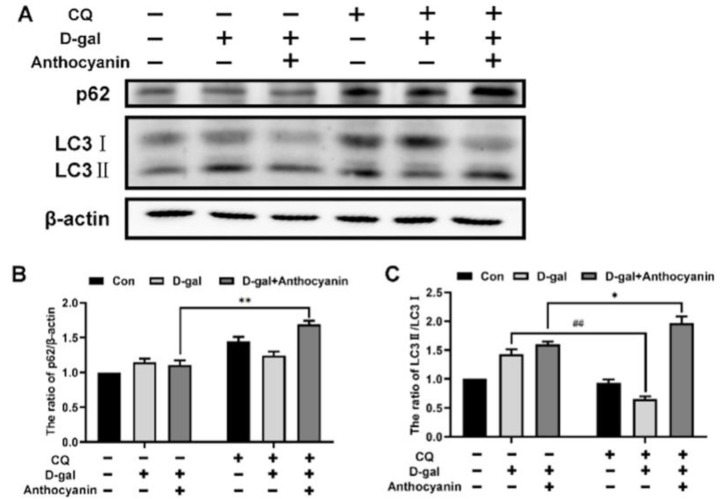
Anthocyanin increased autophagic flux in senescent cells induced by D-galactose. (**A**) Cells were treated with 40 mM D-galactose alone or in combination with 40 μg/mL anthocyanin and 10 μg/mL CQ for 72 h, and the expressions of p62, LC3II/I were measured using western blot. (**B**,**C**) The quantification of protein expressions. Data are presented as mean ± SD, n = 3. * *p* < 0.05, ** *p* < 0.01, ^##^
*p* < 0.01.

**Figure 7 ijms-24-01528-f007:**
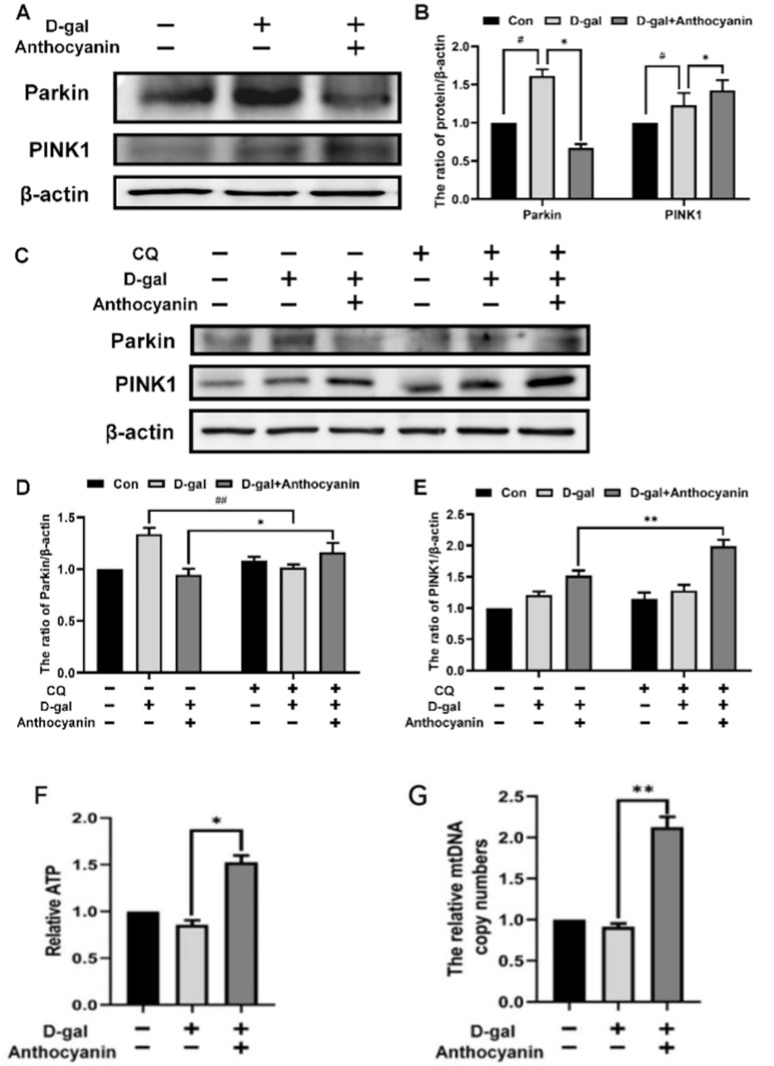
Anthocyanin improved the mitochondrial function of senescent cells. Cells were treated with 40 mM D-galactose alone or in combination with 80 μg/mL anthocyanin and CQ for 72 h. (**A**–**E**) The expressions of PINK1 and Parkin were measured using western blot. (**F**) The analysis of the intracellular ATP concentration. (**G**) The relative mtDNA copy numbers. Data are presented as mean ± SD, n = 3. * *p* < 0.05, ** *p* < 0.01, ^#^
*p* < 0.05, ^##^
*p* < 0.01.

**Figure 8 ijms-24-01528-f008:**
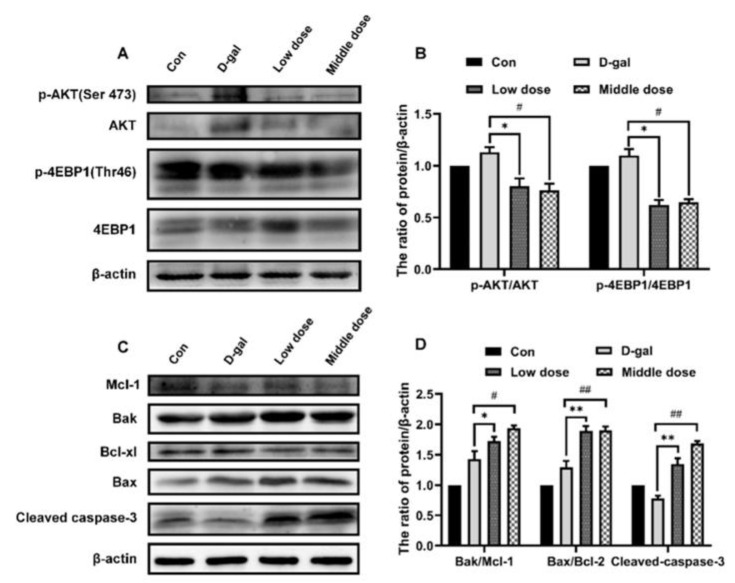
Anthocyanin decreased the activity of PI3K/AKT/mTOR and increased apoptosis in mice lenses induced by D-galactose. (**A**,**B**) The expression of p-AKT/AKT, p-4EBP1/4EBP1 were measured using western blot. (**C**,**D**) The expression of Bcl-2 family proteins and Cleaved-caspase-3 were measured using western blot. Data are presented as mean ± SD, n = 3. * *p* < 0.05, ** *p* < 0.01, ^#^
*p* < 0.05, ^##^
*p* < 0.01.

**Figure 9 ijms-24-01528-f009:**
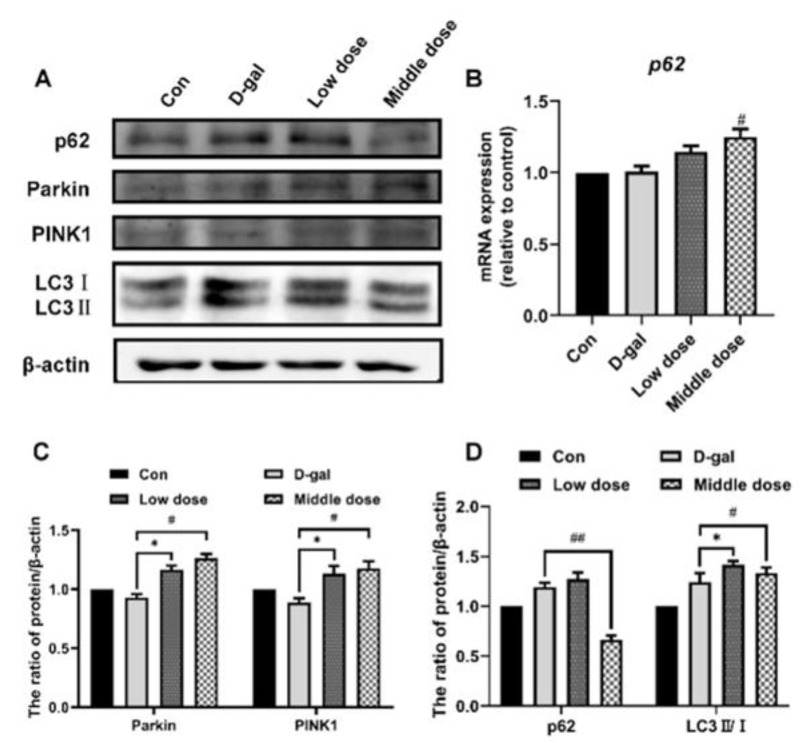
Anthocyanin increased the autophagic flux of mice lenses. (**A**) The expression of p62, LC3II/I, PINK1, and Parkin were measured using western blot. (**B**) The mRNA expression of p62 was measured using RT-qPCR. (**C**,**D**). The quantification of expression of p62, LC3II/I, PINK1, and Parkin. Data are presented as mean ± SD, n = 3. * *p* < 0.05, ^#^
*p* < 0.05, ^##^
*p* < 0.01.

**Figure 10 ijms-24-01528-f010:**
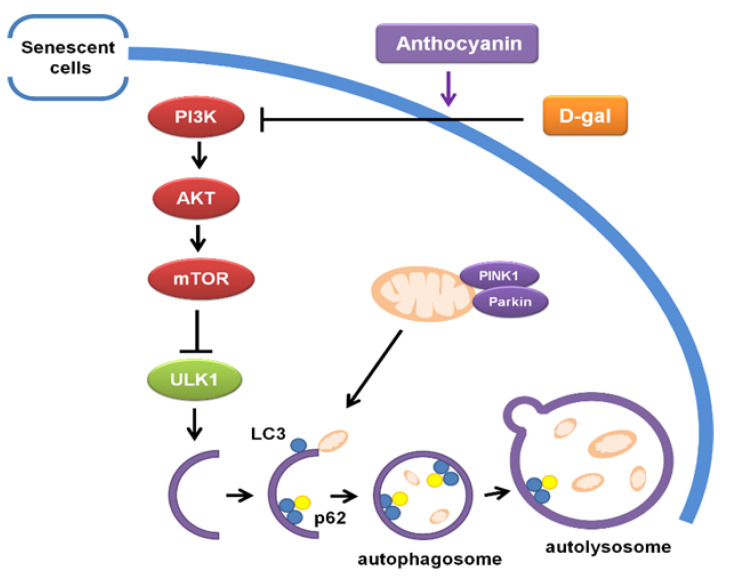
Proposed mechanism schema for Anthocyanin attenuates aging through clearing senescent cells by inducing apoptosis and autophagy via the inactivation of PI3K/AKT/mTOR pathway. Anthocyanin enhanced autophagic and mitophagic flux in senescent cells through inhibiting the activity of PI3K/AKT/mTOR pathway, clearing damaged organelles such as mitochondria, and promoted apoptosis of senescent cells to attenuate aging.

**Table 1 ijms-24-01528-t001:** The results of hematologic examination.

	Con	D-gal	Low Dose	Middle Dose	High Dose	VE
Glucose (mmol/L)	8.16 ± 0.27	9.37 ± 0.32	6.14 ± 0.21	7.95 ± 0.19	7.29 ± 0.23	7.25 ± 0.25
Aspartate Aminotransferase (U/L)	138.7 ± 3.5	158.5 ± 4.9	140.3 ± 5.3	122.6 ± 4.2	221.9 ± 6.7	229.5 ± 7.2
Total protein (g/L)	61.7 ± 0.73	52.0 ± 0.85	56.1 ± 0.65	59.9 ± 0.77	59.4 ± 0.69	61.2 ± 0.58
Albumin (g/L)	31.3 ± 0.26	24.3 ± 0.28	26.3 ± 0.32	32.9 ± 0.29	27.9 ± 0.38	29.3 ± 0.24

## Data Availability

Data sharing is not applicable.

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
