# Peer review of "Anti-Aging Effects of Anthocyanin Extracts of *Sambucus canadensis* Caused by Targeting Mitochondrial-Induced Oxidative Stress"

_ijms, 2023, doi:10.3390/ijms24021528_

Round 1
Reviewer 1 Report
comments are found in the manuscript

Author Response
#Response to Reviewer 1
1. italic in the manuscript.
Response:
We have checked and modified in the revised manuscript.
2. percentage data of the global or local problem can be mentioned.
Response:
We have modified in the revised manuscript.
3. mention how long this inhibition extends life and whether any adverse conditions were not present.
Response:
We have modified in the revised manuscript.
4. do not use first person plural, rewrite.
Response:
We have rewritten the sentence in the revised manuscript.
5. please mention how much this adverse effect amounts to.
Response:
We have modified in the revised manuscript.
6. put full name of the plant with the order and taxonomic family to which it belongs.
Response:
We have supplied the full name in the revised manuscript.
7. mention how long this inhibition extends life and whether any adverse conditions were not present.
Response:
We have modified in the revised manuscript.
8. What is the meaning of VE, you can put it at the end of the table. What is the meaning of CON,you can put it at the end of the table. you can specify more, remember that the table should stand on its own.
Response:
In this study, VE group was used as a positive control, the CON group was used as a negative control.
9. what is the meaning of MTT? what is the meaning of EtOAc ?mention the meaning of VE. mention the meaning of SDSPAGE. mention the meaning of ATP. mention the meaning of MIT, and support this part with a scientific reference. mention the meaning of PVDF
A sample of 36..............
mention the meaning of SPF
Response:
According to the suggestion, we have added the related abbreviations in the revised manuscript. In addition, we chose the SPF grade Kunming mice for this study. In the early experiments, we compared different species or strain animals in aging related experiments. We found that Kunming mice are more suitable for aging related experiments than BALB/c mice and C57BL/6J mice, probably because they belong to closed colony animals, while c57 belongs to inbred strain animals. We used the Kunming mice in another published paper. (PMID: 35805987) According to the welfare and ethical principles of experimental animals, we used as few mice as possible. It is 6 mice in each group, so there are 36 mice in this study.
10. there is no section on experimental design and statistical analysis, mention it. this methodological aspect needs to be strengthened with some reference
Response:
We have added the data statistical analysis in the materials and methods.
Reviewer 2 Report
The article entitled "Anti-aging effects of anthocyanin extracts of Sambucus canadensis by targeting mitochondrial-induced oxidative stress" shows an anti-aging potential of anthocyanin extracts of Sambucus canadensis. The paper is well written and concise. However, some information needs to be better detailed and fixed for possible publication in [International Journal of Molecular Sciences].
1. English language and style are minor spell check required.
2. How was the filtration and evaporation of the extracts? Detail this methodology.
3. Which article or methodology was used to calculate the sample number of animals used?
4. Item 4.5. (Cell viability assay) needs a better explanation.
5. It is necessary to add a statistical analysis topic in the methodology, as well as insert in each figure caption the statistics used for each experiment to reach the p values mentioned in the work.
Author Response
#Response to Reviewer 2
The article entitled "Anti-aging effects of anthocyanin extracts of Sambucus canadensis by targeting mitochondrial-induced oxidative stress" shows an anti-aging potential of anthocyanin extracts of Sambucus canadensis. The paper is well written and concise. However, some information needs to be better detailed and fixed for possible publication in [International Journal of Molecular Sciences].
1. English language and style are minor spell check required.
Response:
We have checked and modified in the revised manuscript.
2. How was the filtration and evaporation of the extracts? Detail this methodology.
Response:
According to the reviewer’s suggestion, we have supplied the extract in the materials and methods.
3. Which article or methodology was used to calculate the sample number of animals used?
Response:
According to the welfare and ethical principles of experimental animals, we used as few mice as possible. It is 6 mice in each group. In addition, we chose the SPF grade Kunming mice for this study. In the early experiments, we compared different species or strain animals in aging related experiments. We found that Kunming mice are more suitable for aging related experiments than BALB/c mice and C57BL/6J mice, probably because they belong to closed colony animals, while c57 belongs to inbred strain animals. We used the Kunming mice in another published paper. (PMID: 35805987)
4. Item 4.5. (Cell viability assay) needs a better explanation.
Response:
We have supplied the method of MTT assay in the 4.5.
5. It is necessary to add a statistical analysis topic in the methodology, as well as insert in each figure caption the statistics used for each experiment to reach the p values mentioned in the work.
Response:
We have added the data statistical analysis in the materials and methods.
Reviewer 3 Report
1) The quality of the images in Figure 2 panel C needs to be improved. In general, the size and quality of the all figures could be improved.
2) The possibility of using nanoparticles to improve the effectiveness of Anthocyanin and its derivatives should be discussed using the example of other natural antioxidants. https://pubmed.ncbi.nlm.nih.gov/36432668/ https://pubmed.ncbi.nlm.nih.gov/34639150/
3) The conclusion should be written in more detail.
Author Response
#Response to Reviewer 3
1) The quality of the images in Figure 2 panel C needs to be improved. In general, the size and quality of the all figures could be improved.
Response:
Thanks for the reviewer’s suggestion. Because the COVID-19, we couldn’t enter the lab and take the western blot again. We do our best to improve the quality of figures by improve the clarity in Figure 2.
2) The possibility of using nanoparticles to improve the effectiveness of Anthocyanin and its derivatives should be discussed using the example of other natural antioxidants. https://pubmed.ncbi.nlm.nih.gov/36432668/
https://pubmed.ncbi.nlm.nih.gov/34639150/
Response:
Thanks for the reviewer’s good suggestion. We have added the related discussion in the revised manuscript.
3) The conclusion should be written in more detail.
Response:
According to the advice, we have rewritten the conclusion in the revised manuscript.